

# Efficient facial representations for age, gender and identity recognition in organizing photo albums using multi-output ConvNet

Andrey V. Savchenko[1,2]

[1] National Research University Higher School of Economics, Laboratory of Algorithms and Technologies for Network Analysis, Nizhny Novgorod, Russia
[2] Samsung-PDMI Joint AI Center, St. Petersburg Department of Steklov Institute of Mathematics, St. Petersburg, Russia

## ABSTRACT

This paper is focused on the automatic extraction of persons and their attributes (gender, year of born) from album of photos and videos. A two-stage approach is proposed in which, firstly, the convolutional neural network simultaneously predicts age/gender from all photos and additionally extracts facial representations suitable for face identification. Here the MobileNet is modified and is preliminarily trained to perform face recognition in order to additionally recognize age and gender. The age is estimated as the expected value of top predictions in the neural network. In the second stage of the proposed approach, extracted faces are grouped using hierarchical agglomerative clustering techniques. The birth year and gender of a person in each cluster are estimated using aggregation of predictions for individual photos. The proposed approach is implemented in an Android mobile application. It is experimentally demonstrated that the quality of facial clustering for the developed network is competitive with the state-of-the-art results achieved by deep neural networks, though implementation of the proposed approach is much computationally cheaper. Moreover, this approach is characterized by more accurate age/gender recognition when compared to the publicly available models.

## INTRODUCTION

Nowadays, due to the extreme increase in multimedia resources, there is an urgent need to develop intelligent methods to process and organize them (*Manju & Valarmathie, 2015*). For example, the task of automatic structuring of photo and video albums is attracting increasing attention (*Sokolova, Kharchevnikova & Savchenko, 2017*; *Zhang & Lu, 2002*). The various photo organizing systems allow users to group and tag photos and videos in order to retrieve large number of images in the media library (*He et al., 2017*). The most typical processing of a gallery includes the faces grouping with automatic assignments of tags with the facial attributes (e.g., age and gender) to each subject in a group. The task of this paper is formulated as follows: given a large number of unlabeled

Corresponding author
Andrey V. Savchenko,
avsavchenko@hse.ru

facial images, cluster the images into individual persons (identities) (*He et al., 2017*) and predict age and gender of each person (*Rothe, Timofte & Van Gool, 2015*).

This problem is usually solved using deep convolutional neural networks (ConvNets) (*Goodfellow, Bengio & Courville, 2016*). At first, the clustering of photos and videos that contains the same person is performed using the known face verification (*Crosswhite et al., 2017*; *Wang et al., 2018*) and identification (*Savchenko & Belova, 2018*) methods. The age and gender of extracted faces can be recognized by other ConvNets (*Eidinger, Enbar & Hassner, 2014*; *Rothe, Timofte & Van Gool, 2015*). Though such an approach works rather well, it requires at least three different ConvNets, which increases the processing time, especially if the gallery should be organized on mobile platforms in offline mode. Moreover, every ConvNet learns its own face representation, and the quality can be limited by the small size of the training set or the noise in the training data. For example, the latter issue is especially crucial for age prediction, as the most widely used IMDB-Wiki dataset contains incorrect ground truth values of age due to mistakes in the year when the photo was taken. Such mistakes are introduced by an automatic crawling procedure used by *Rothe, Timofte & Van Gool (2015)*.

Therefore, the goal of this research is to improve the efficiency of facial clustering and age and gender prediction by learning face representation using preliminarily training on domain of unconstrained face identification from very large database. The contribution of this paper can be formulated as follows. Firstly, a multi-output extension of the MobileNet (*Howard et al., 2017*) is specially developed. It is pre-trained to perform face recognition using the VGGFace2 dataset (*Cao et al., 2018*). Additional layers of the proposed network are fine-tuned for age and gender recognition on Adience (*Eidinger, Enbar & Hassner, 2014*) and IMDB-Wiki (*Rothe, Timofte & Van Gool, 2015*) datasets. Secondly, it is proposed to estimate the age of the person by computing the expected value of top predictions in the age head of the proposed neural network. Finally, a novel approach to face grouping is proposed, which deals with several challenges of processing of real-world photo and video albums.

## RELATED WORKS

### Face clustering

Contemporary deep learning techniques (*Goodfellow, Bengio & Courville, 2016*) can deal even with well-known crucial issues appeared in practical applications of face recognition, for example, unconstrained environment (various illumination and pose, partial occlusion) (*Learned-Miller et al., 2016*), or the small sample size problem (*Savchenko, 2016*), when usually only single facial image per person is available. The latter problem is solved using transfer learning or domain adaptation methods (*Goodfellow, Bengio & Courville, 2016*), in which large external datasets of celebrities are used to train deep ConvNet (*Cao et al., 2018*; *Parkhi, Vedaldi & Zisserman, 2015*).

One of the main tasks of this paper is to group (cluster) the images into individual identities present in the given data with a large number of face images (*He et al., 2017*). In this task transfer learning techniques with fine-tuning of the ConvNet into the new training set of persons of interest is impossible because the images are unlabeled; that is,

the setting is completely unsupervised. Hence, traditional face clustering methods focus on finding effective face representation or appropriate dissimilarity measure between faces. The latter approach is examined by *Zhu, Wen & Sun (2011)*, who proposed a rank-order distance that measures the similarity between two faces using their neighboring information. *Shi, Otto & Jain (2018)* designed a clustering algorithm, which directly estimates the adjacency matrix only based on the similarities between face images. This allows a dynamic selection of number of clusters and retains pairwise similarities between faces. However, the most accurate grouping in many practical tasks is still achieved by the usage of reliable face representations with consecutive agglomerative clustering (*Zhang et al., 2016b*). In this case, the ConvNet is pre-trained on external large dataset and is further applied to extract features of the training images from the limited sample of subjects using embeddings at one of the last layers (*Sharif Razavian et al., 2014*; *Savchenko, 2019*).

Extraction of representative features (embeddings) is one the most important tasks in face recognition. *Taigman et al. (2014)* provided one of the first implementations of ConvNets for face verification using the DeepFace descriptor trained with simple negative log-likelihood ("softmax") loss. Various regularizations of the loss functions have been studied in order to provide suitable representations. For example, center loss has been proposed by *Wen et al. (2016)* in order to simultaneously learn a center for deep features of each class and penalize the distances between the deep features and their corresponding class centers. *Guo & Zhang (2017)* proposed underrepresented-classes promotion loss term, which aligns the norms of the weight vectors of the underrepresented-classes to those of the normal classes. The Deep IDentification-verification features (DeepID2) are learned by alternating identification and verification phases with different loss functions (*Sun et al., 2014*). FaceNet descriptors (*Schroff, Kalenichenko & Philbin, 2015*) were trained using special triplet loss with triplets of roughly aligned matching/non-matching face patches. Recently, a family of angular and cosine margin-based losses have appeared. For example, the angular softmax loss was used to learn SphereFace descriptors (*Liu et al., 2017b*). The ArcFace loss (*Deng et al., 2019*) directly maximizes decision boundary in angular (arc) space based on the $L_2$-normalized weights and features.

However, it is worth noting that the usage of softmax loss still gives the most accurate representations if the dataset is very large. *Cao et al. (2018)* gathered VGGFace-2 dataset with 3M photos of 10K subjects and trained conventional ResNet-based networks, which achieve state-of-the-art results in various face recognition tasks. Recently, the research directions have been shifted into learning a compact embedding using ConvNets with low latency. For example, *Wu et al. (2018)* introduced the concept of maxout activation and proposed a light ConvNet suitable for fast but still accurate face recognition.

## Facial attributes recognition

Recognition of facial attributes appeared in many practical applications. For instance, one of the goals of video analysis in retail is to fill the advertisements with relevant information, interesting to a target group. In this paper, it was decided to focus on age and gender

(*Kharchevnikova & Savchenko, 2016*), which are the most important attributes in the automatic organization of a gallery of photos.

A decade ago traditional computer vision techniques, for example, classification of Gabor features have been thoroughly studied (*Choi et al., 2011*). Nowadays, the most prominent results are achieved by deep ConvNets. For example, *Eidinger, Enbar & Hassner (2014)* gather the Adience dataset and trained the Gender_net and Age_net models, which achieved very accurate classification. Also, *Rothe, Timofte & Van Gool (2015)* provided large IMDB-Wiki dataset and trained deep VGG-16 ConvNets, which achieved the state-of-the-art results in gender and age recognition. *Antipov et al. (2017)* extended the paper (*Rothe, Timofte & Van Gool, 2015*) of and demonstrated that the transfer learning with the face recognition pre-training (*Rassadin, Gruzdev & Savchenko, 2017*) is more effective for gender and age recognition compared to general pre-training using ImageNet dataset.

Unfortunately, the above-mentioned ConvNet-based methods are characterized by considerable memory consumption and computational complexity. More importantly, the size of the datasets with labeled age and gender attributes is rather low when compared to the datasets used for face recognition mentioned in the previous subsection. It is rather obvious that the closeness among the facial processing tasks can be exploited in order to learn efficient face representations which boosts up their individual performances. Hence, one of the main parts of this paper is to use the embeddings trained for face identification in order to predict age and gender of a given person.

There exist several studies, which use such multi-task approach. For instance, *Han et al. (2018)* trained a single ConvNet to classify several facial attributes. Simultaneous face detection, landmark localization, pose estimation, and gender recognition is implemented in the paper (*Ranjan, Patel & Chellappa, 2017*) by a single ConvNet. However, this model is primarily focused only on rather simple task of localization of a facial region (face detection). It does not extract useful facial representations (features) and cannot be used in face identification and/or clustering.

Another important example of simultaneous facial analysis is presented in the patent application (*Yoo et al., 2018*). The task solved by this invention is rather close to the task considered in this paper, namely, prediction of identifier (ID) and human attributes (age, gender, etc.) given a facial image using multi-task training. However, this device does not predict the birth year as it only classifies the age range, that is, "baby" (0–7 years), "child" (8–12), "senior" (61-...), etc. Secondly, all the recognition tasks (ID and facial attributes) are trained simultaneously in this invention, which requires the training set to include facial images with known ID and all attributes. In practice, the existing face datasets do not include all this information. Hence, this method restricts the used training set, and, as a consequence, cannot provide the highest accuracies in all tasks. Finally, this model predicts the ID from a limited set of IDs from the training set. Hence, it cannot be used for face identification with small training samples because it does not implement the domain adaptation and is restricted to the IDs from the given training set. More importantly it is impossible to apply this method for organizing the photo albums in completely unsupervised environment with unknown persons.
It seems that there is still a lack of studies devoted to simultaneous extraction of reliable representations and face attribute classification suitable for face grouping and age/gender prediction for each person using a set of his or her photos.

## MULTI-OUTPUT CONVNET FOR SIMULTANEOUS AGE, GENDER, AND IDENTITY RECOGNITION

In this paper, several different facial analytic tasks are considered. It is assumed that the facial regions are obtained in each image using any appropriate face detector; for example, either traditional multi-view cascade Viola–Jones classifier or more accurate ConvNet-based methods (*Zhang et al., 2016a*). The *gender* recognition task is a binary *classification* problem, in which the obtained facial image is assigned to one of two classes (male and female). The *age* prediction is the special case of *regression* problem, though sometimes it is considered as a multi-class classification with, for example, $N = 100$ different classes, so that it is required to predict if an observed person is $1, 2, \ldots$ or 100 years old (*Rothe, Timofte & Van Gool, 2015*). In such a case, these two tasks become very similar and can be solved by traditional deep learning techniques. Namely, the large facial dataset of persons with known age and/or gender is gathered; for example, the IMDB-Wiki (*Rothe, Timofte & Van Gool, 2015*). After that the deep ConvNet is learned to solve the classification task. The resulted networks can be applied to predict age and gender given a new facial image.

The last task examined in this paper, namely, unconstrained *face identification* significantly differs from age and gender recognition. The unsupervised learning case is considered, in which facial images from a gallery set should be assigned to one of $C \geq 1$ subjects (identities). Domain adaptation (*Goodfellow, Bengio & Courville, 2016*) is usually applied here: each image is described with the off-the-shelf feature vector using the deep ConvNet (*Sharif Razavian et al., 2014*), which has been preliminarily trained for the *supervised face identification* on large external dataset, for example, CASIA-WebFace, VGGFace/VGGFace2, or MS-Celeb-1M. The $L_2$-normalized outputs at the one of last layers of this ConvNet for each $r$th gallery image are used as the $D$-dimensional feature vectors $\mathbf{x}_r = [x_{r;1}, \ldots, x_{r;D}]$. Finally, any appropriate clustering method, that is, hierarchical agglomerative clustering (*Aggarwal & Reddy, 2013*), can be used to make a final decision for these feature vectors.

In most research studies all these tasks are solved by independent ConvNets even though it is necessary to solve all of them. As a result, the processing of each facial image becomes time-consuming, especially for offline mobile applications (*Kharchevnikova & Savchenko, 2018*). In this paper it is proposed to solve all these tasks by the same ConvNet. In particular, it is assumed that the features extracted during face identification can be rather rich for any facial analysis. For example, it was shown the VGGFace features (*Parkhi, Vedaldi & Zisserman, 2015*) can be used to increase the accuracy of visual emotion recognition (*Kaya, Gürpnar & Salah, 2017*; *Rassadin, Gruzdev & Savchenko, 2017*). As the main requirement in this study is the possibility to use the ConvNet on mobile platforms, it was decided to use straightforward modification of the MobileNet v1 (*Howard et al., 2017*) (Fig. 1). This model contains 27 sequentially connected

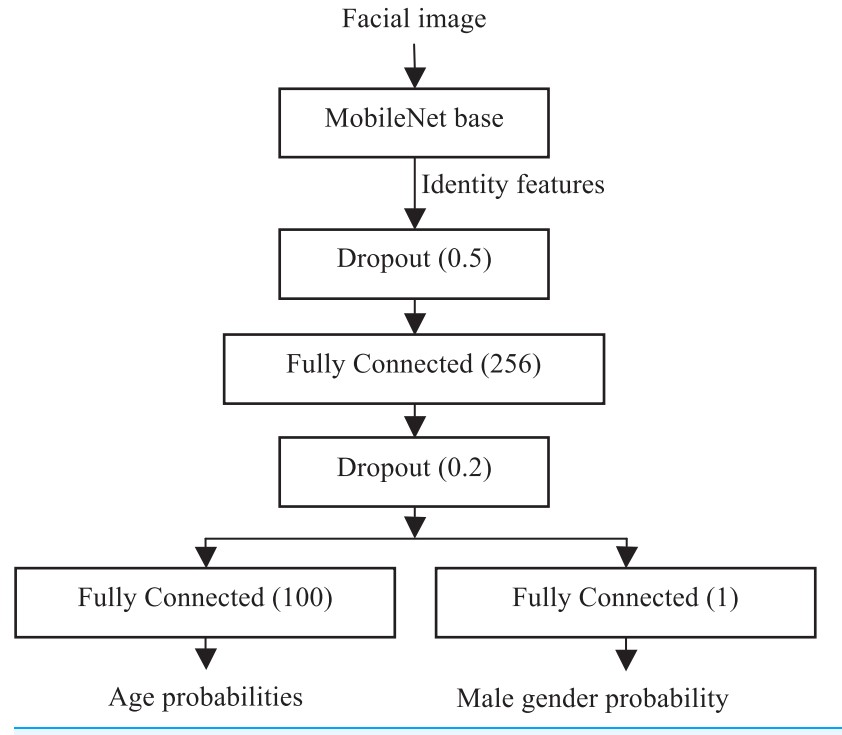

**Figure 1 Proposed ConvNet.**

convolutional and depthwise convolutional layers, which proved to be memory efficient and provide excellent inference speed even on mobile devices. It is seven- and 35-times smaller than conventional ResNet-50 and VGG16 models, respectively. What is more important, such small size of the model does not cause significant decrease of the recognition accuracy in various image recognition tasks. For example, top-1 accuracy on ImageNet-1000 of the MobileNet v1 (70.4%) is only 0.9% and 4.5% lower when compared to the accuracy of VGG16 and ResNet-50, respectively.

The bottom (backbone) part of the proposed network, namely, conventional MobileNet v1 pre-trained on ImageNet-1000, extracts the representations suitable for face identification. The top part contains one new hidden layer with dropout regularization after extraction of identity features and two independent fully connected layers with softmax and sigmoid outputs for age and gender recognition, respectively. The learning of this model is performed incrementally, At first, the base MobileNet is trained for face identification on a very large facial dataset using conventional cross-entropy (softmax) loss. Next, the last classification layer is removed, and the weights of the MobileNet base are frozen. Finally, the remaining layers in the head are learned for age and gender recognition tasks by minimizing the sum of cross-entropies for both outputs.

It is necessary to emphasize that not all images in most datasets of facial attributes contain information about both age and gender. Moreover, some attribute may be completely unknown, if several datasets are united. As a result, the number of faces with both age and gender information is several times smaller when compared to the whole number of facial images. Finally, the gender data for different ages is also very imbalanced.

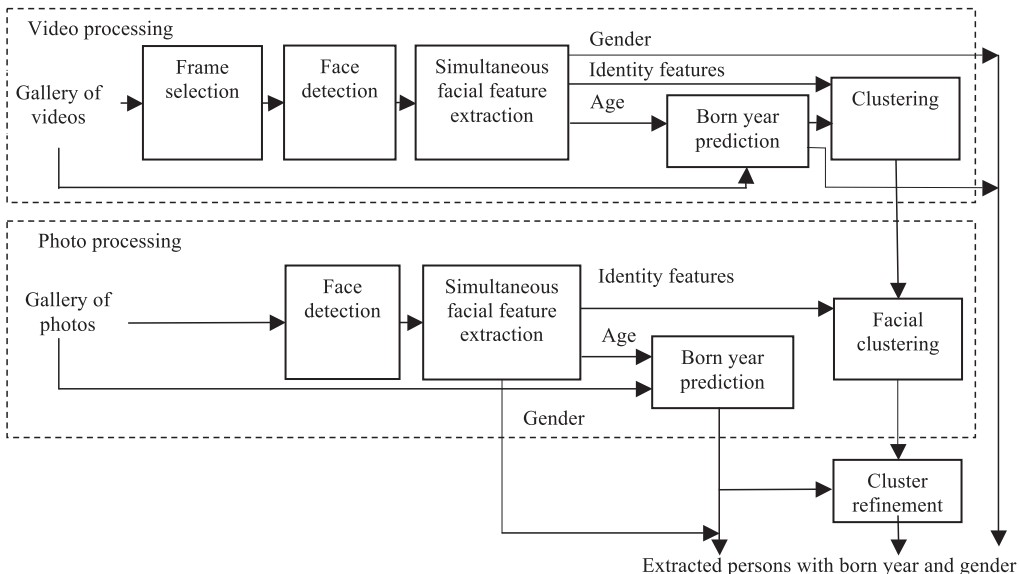

**Figure 2  Proposed pipeline.**

Thus, it was decided to train both heads of the ConvNet (Fig. 1) independently using different training data for age and gender classification. In particular, I alternate the mini-batches with age and gender info, and train only the part of the proposed network, that is, the weights of the fully connected layer in the age head of the proposed model are not updated for the mini-batch with the gender info.

This ConvNet has the following advantages. First of all, it is obviously very efficient due to either usage of the MobileNet backbone or the possibility to simultaneously solve all three tasks (age, gender, and identity recognition) without need to implement an inference in three different networks. Secondly, in contrast to the publicly available datasets for age and gender prediction, which are rather small (compared to the datasets for face recognition) and dirty, the proposed model exploit the potential of very large and clean face identification datasets to learn very good face representation. Moreover, the hidden layer between the identity features and two outputs further combines the knowledge necessary to predict age and gender. As a result, the proposed model makes it possible to increase the accuracy of age/gender recognition when compared to the models trained only on specific datasets; for example, IMDB-Wiki or Adience.

## PROPOSED PIPELINE FOR ORGANIZING PHOTO AND VIDEO ALBUMS

The complete data flow of the usage of the ConvNet (Fig. 1) for organizing albums with photos and videos the is presented in Fig. 2. Here faces in each photo are detected using the MTCNN (multi-task convolutional neural network) (Zhang et al., 2016a). Next, an inference in the proposed ConvNet is performed for all detected faces $X_r$ in order to extract D-dimensional identity feature vector $\mathbf{x}_r$ and predict age and gender. Current age $a_r$ of the $r$th person is estimated by adding a difference between current date and the photo creation date to the predicted age. After that, all faces are clustered using the following

dissimilarity measure between identity features and birth year predictions of two facial images $X_r$ and $X_j$:

$$\rho(X_r, X_j) = \parallel \mathbf{x}_r - \mathbf{x}_j \parallel_2^2 + w \frac{(a_r - a_j)^2}{a_r + a_j}, \qquad (1)$$

where $\|\mathbf{x}\|_2$ is $L_2$ norm and $w \geq 0$ is a fixed weight (e.g., 0.1) of linear scalarization. The second term of (1) was chosen for age distance in order to maximize the difference between small babies, who usually have similar identity features.

As the number of subjects in the photo albums is usually unknown, a hierarchical agglomerative clustering is used (*Aggarwal & Reddy, 2013*). Only rather large clusters with a minimal number of faces are retained during the cluster refinement. The gender and the birth year of a person in each cluster are estimated by appropriate fusion techniques (*Kharchevnikova & Savchenko, 2016*, *2018*); for example, simple voting or maximizing the average posterior probabilities at the output of the ConvNet (Fig. 1). For example, the product rule (*Kittler et al., 1998*) can be applied if the independence of all facial images $X_r$, $r \in \{r_1, \ldots, r_M\}$ in a cluster is naively assumed:

$$\max_{n \in \{1, \ldots, N\}} \prod_{m=1}^{M} p_n(X_{r_m}) = \max_{n \in \{1, \ldots, N\}} \sum_{m=1}^{M} \log p_n(X_{r_m}), \qquad (2)$$

where $N$ is the total number of classes and $p_n(X_{rm})$ is the $n$th output of the ConvNet for the input image $X_{rm}$.

The same procedure is repeated for all video files. Only each of, for example, three or five frames, is selected in each video clip, extract identity features of all detected faces and initially cluster *only* the faces found in this clip. After that the normalized average of identity features of all clusters (*Sokolova, Kharchevnikova & Savchenko, 2017*) are computed. They are added to the dataset $\{X_r\}$ so that the "Facial clustering" module handles both features of all photos and average feature vectors of subjects found in all videos.

Let me summarize the main novel parts of this pipeline. Firstly, the simple age prediction by maximizing the output of the corresponding head of the proposed ConvNet is not accurate due to the imbalance of described training set, which leads to the decision in favor of one of the majority class. Hence, it was decided to aggregate the outputs $\{p_a(X_r)\}$ of the age head. However, I experimentally found that the combination of *all* outputs is again inaccurate, because the majority of subjects in the training set are 20–40 years old. Thus, it is proposed to choose only $L \in \{1, 2, \ldots, N_a\}$ indices $\{a_1, \ldots, a_L\}$ of the maximal outputs, where $N_a$ is the number of different age classes. Next, the expected mean $\bar{a}(X_r)$ for each gallery facial image $X_r$ is computed using normalized top outputs as follows:

$$\bar{a}(X_r) = \frac{\sum\limits_{l=1}^{L} a_l \cdot p_{al}(X_r)}{\sum\limits_{l=1}^{L} p_{al}(X_r)} \qquad (3)$$

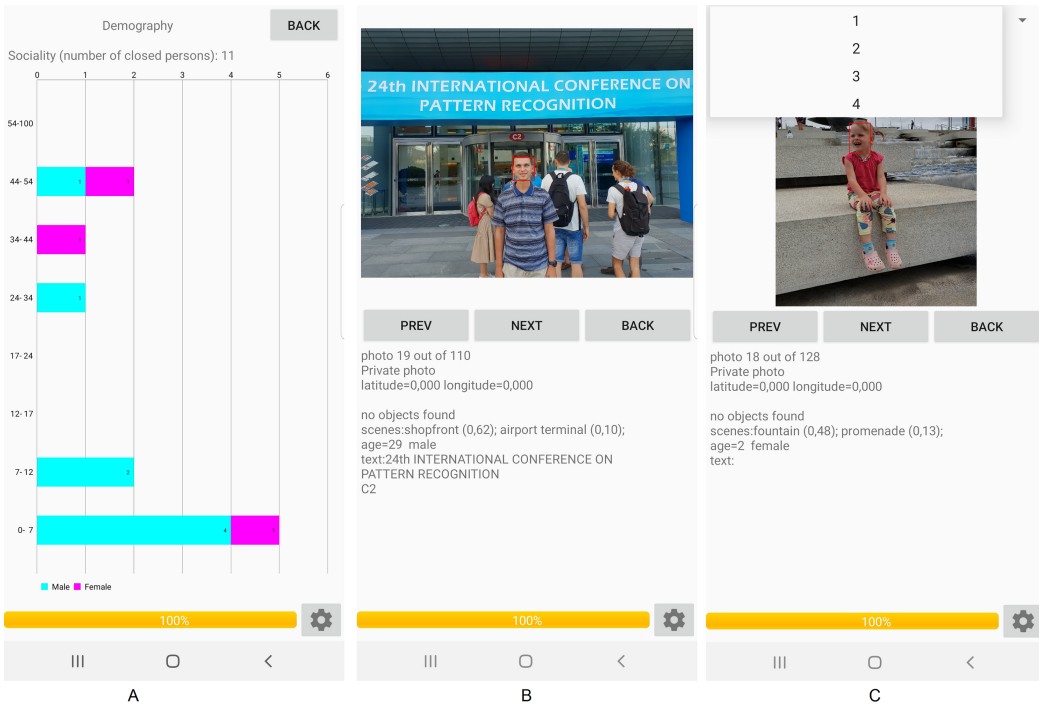

**Figure 3 Results of organization of the gallery in the mobile demo.** Photo credit: Andrey V. Savchenko.

Secondly, the birth year of each face is estimated by subtracting the predicted age from the image file creation date. In such a case, it will be possible to organize the very large albums gathered over the years. The predicted birth year is used as an additional feature during the cluster analysis in order to partially overcome the known similarity of young babies in a family. The distance between two faces are computed as a sum of a distance between facial features and appropriately scaled distance (1) between predicted current ages with respect to the photo creation dates.

Finally, several tricks in the cluster refinement module were implemented (Fig. 2). At first, the different faces appeared on the same photo are specially marked. As such faces must be stored in different groups, complete linkage clustering of every facial cluster is additionally performed. The distance matrix is designed so that the distances between the faces at the same photo are set to the maximum value, which is much larger than the threshold applied when forming flat clusters. Moreover, it is assumed that the most valuable clusters of an owner of mobile device, his or her friends and relatives should not contain photos/videos taken in only one day. Hence, a certain threshold (1 day by default) for a number of days between the earliest and the eldest photo in a cluster is set in order to disambiguate large quantity of unimportant causal faces.

The proposed approach (Fig. 2) was implemented as a part of a special mobile application for Android (Fig. 3). The application may operate in offline mode and does not require an Internet connection. It sequentially processes all photos from the gallery in a background thread. The demography pane provides stacked histograms (Fig. 3A) of facial attributes of the largest extracted clusters (family members and friends). Tapping on

each bar within the horizontal stacked histogram in Fig. 3A causes the list of all photos of a particular individual to be displayed (Fig. 3B). It is important to emphasize at this point that entire photos rather than just faces extracted therefrom are presented in the display form of the application, so that photos with several persons can be exposed. If there are plural individuals with an identical gender and age range, then a spinner (combo box) can be provided on top of the display form, and that spinner is usable to select a particular person by an associated sequential number (Fig. 3C).

# EXPERIMENTAL RESULTS

## Details of the training procedure

In order to test the described approach (Fig. 2), I implemented it in a special software (https://github.com/HSE-asavchenko/HSE_FaceRec_tf/tree/master/age_gender_identity) using Python language with the Tensorflow and Keras frameworks and the Scikit-learn/ SciPy/NumPy libraries. My fine-tuned CNNs are publicly available. The network (Fig. 1) has been trained as follows. At first, the weights of the backbone MobileNet are learned for face identification problem using 3,067,564 facial images of 9,132 subjects from VGGFace2 dataset (*Cao et al., 2018*). This base CNN was trained for 20 epochs with early stopping on validation set of 243,722 other images using Adam optimizer of softmax loss with learning rate 0.001 and decay 1$E$-5.

Next, I populated the training dataset by 300K frontal cropped facial images from the IMDB-Wiki dataset (*Rothe, Timofte & Van Gool, 2015*). Unfortunately, the age groups in this dataset are very imbalanced, so the trained models work incorrectly for faces of very young or old people. Hence, it was decided to add all (15K) images from the Adience (*Eidinger, Enbar & Hassner, 2014*) dataset. As the latter contains only age intervals, for example, "(0–2)," "(60–100)," I put all images from this interval to the middle age, for example, "1" or "80." The resulted training set contains partially overlapped 22,084 images with gender label and 216,465 images with age label. The validation set includes other 26,230 labeled images with both age and gender available.

In this study both age and gender recognition tasks are treated as classification problems with $N_g = 2$ (male/female) and (1,2,...100 years old) classes, respectively. The proposed network is fine-tuned on the gathered dataset in order to minimize the following joint loss function

$$L = -\sum_{g \in G} \log p_{n_g}(X_i) - \sum_{a \in A} \log p_{n_a}(X_i), \tag{4}$$

which is computed as a sum of gender binary cross-entropy loss and age categorical cross-entropy loss. Here $p_n(X)$ is the $n$th output of the ConvNet for the input image $X$, that is, the estimate of posterior probability of the $n$th class, $G$ is a set of indices $g$ of images in the training set with known gender label $n_g$. Similarly, set $A$ contains indices $a$ of training images with given age $n_a$. In order to compute the above-mentioned cross entropy loss functions, one-hot encoding is used for both age and gender labels.

The top part of the network (Fig. 1) with frozen weights of the base CNN has been learned for three epochs using Adam optimizer of the loss (4) with alternate age/gender

batches. As a result, 97% and 13% validation accuracies were obtained for age and gender, respectively. If the whole network including the backbone MobileNet is fine-tuned for one epoch using the SGD optimizer with learning rate $1E$-4, these accuracies are increased to 98% and 16%, but the quality of identity features suitable for face recognition obviously decreases.

## Facial clustering

In this subsection experimental study of the proposed system (Fig. 2) are provided in facial clustering task for images gathered in unconstrained environments. The identity features extracted by the base MobileNet (Fig. 1) are compared to the publicly available ConvNets suitable for face recognition, namely, the VGGFace (VGGNet-16) (*Parkhi, Vedaldi & Zisserman, 2015*) and the VGGFace2 (ResNet-50) (*Cao et al., 2018*). The VGGFace, VGGFace2, and MobileNet extract $D = 4,096$, $D = 2,048$, and $D = 1,024$ non-negative features in the output of "fc7," "pool5_7x7_s1," and "reshape_1/Mean" layers from $224 \times 224$ RGB images, respectively.

All hierarchical clustering methods from SciPy library are used with the Euclidean ($L_2$) distance between feature vectors. As the centroid and the Ward's linkage showed very poor performance in all cases, only results for single, average, complete, weighted, and median linkage methods are reported. In addition, the rank-order clustering (*Zhu, Wen & Sun, 2011*) was implemented, which was specially developed for organizing faces in photo albums. The parameters of all clustering methods were tuned using 10% of each dataset. The following clustering metrics were estimated with the Scikit-learn library: adjusted Rand index, adjusted mutual information, homogeneity, and completeness. In addition, the average number of extracted clusters $K$ relative to the number of subjects $C$ and the BCubed $F$-measure are computed. The latter metric is widely applied in various tasks of grouping faces (*He et al., 2017*; *Zhang et al., 2016b*).

In the experiments the following testing data were used.

- Subset of labeled faces in the wild (LFW) dataset (*Learned-Miller et al., 2016*), which was involved into the face identification protocol (*Best-Rowden et al., 2014*). $C = 596$ subjects who have at least two images in the LFW database and at least one video in the YouTube Faces (YTF) database (subjects in YTF are a subset of those in LFW) are used in all clustering methods.

- Gallagher collection person dataset (*Gallagher & Chen, 2008*), which contains 931 labeled faces with $C = 32$ identities in each of the 589 images. As only eyes positions are available in this dataset, I preliminarily detect faces using MTCNN (*Zhang et al., 2016a*) and chose the subject with the largest intersection of facial region with given eyes region. If the face is not detected, a square region with the size chosen as a 1.5-times distance between eyes is extracted.

- Grouping faces in the wild (GFW) (*He et al., 2017*) with preliminarily detected facial images from 60 real users' albums from a Chinese social network portal. The size of an album varies from 120 to 3,600 faces, with a maximum number of identities of $C = 321$.

**Table 1 Clustering results, LFW subset (*C* = 596 subjects).**

|  |  | *K/C* | ARI | AMI | Homogeneity | Completeness | *F*-measure |
|---|---|---|---|---|---|---|---|
| Single | VGGFace | 1.85 | 0.884 | 0.862 | 0.966 | 0.939 | 0.860 |
|  | VGGFace2 | 1.22 | 0.993 | 0.969 | 0.995 | 0.986 | 0.967 |
|  | Proposed model | 2.00 | 0.983 | 0.851 | 0.998 | 0.935 | 0.880 |
| Average | VGGFace | 1.17 | 0.980 | 0.937 | 0.985 | 0.971 | 0.950 |
|  | VGGFace2 | 1.06 | **0.997** | **0.987** | 0.998 | **0.994** | **0.987** |
|  | Proposed model | 1.11 | 0.995 | 0.971 | 0.993 | 0.987 | 0.966 |
| Complete | VGGFace | 0.88 | 0.616 | 0.848 | 0.962 | 0.929 | 0.823 |
|  | VGGFace2 | 0.91 | 0.760 | 0.952 | 0.986 | 0.978 | 0.932 |
|  | Proposed model | 0.81 | 0.987 | 0.929 | 0.966 | 0.986 | 0.916 |
| Weighted | VGGFace | 1.08 | 0.938 | 0.928 | 0.979 | 0.967 | 0.915 |
|  | VGGFace2 | 1.08 | **0.997** | 0.982 | 0.998 | 0.992 | 0.983 |
|  | Proposed model | 1.08 | 0.969 | 0.959 | 0.990 | 0.981 | 0.986 |
| Median | VGGFace | 2.84 | 0.827 | 0.674 | 0.987 | 0.864 | 0.751 |
|  | VGGFace2 | 1.42 | 0.988 | 0.938 | 0.997 | 0.972 | 0.947 |
|  | Proposed model | 2.73 | 0.932 | 0.724 | **0.999** | 0.884 | 0.791 |
| Rank-order | VGGFace | 0.84 | 0.786 | 0.812 | 0.955 | 0.915 | 0.842 |
|  | VGGFace2 | **0.98** | 0.712 | 0.791 | 0.989 | 0.907 | 0.888 |
|  | Proposed model | 0.86 | 0.766 | 0.810 | 0.962 | 0.915 | 0.863 |

**Note:**
The best results in each column are marked in bold.

The average values of clustering performance metrics are presented in Tables 1–3 for LFW, Gallagher, and GFW datasets, respectively.

The average linkage is the best method according to most of the metrics of cluster analysis. The usage of the rank-order distance (*Zhu, Wen & Sun, 2011*) is not appropriate due to rather low performance. Moreover, this distance requires an additional threshold parameter for the cluster-level rank-order distance. Finally, the computational complexity of such clustering is three to four times higher when compared to other hierarchical agglomerative clustering methods. One of the most important conclusion here is that the trained MobileNet (Fig. 1) is in most cases more accurate than the widely used VGGFace. As expected, the quality of the proposed model is slightly lower when compared to the deep ResNet-50 ConvNet trained on the same VGGFace2 dataset. Surprisingly, the highest BCubed *F*-measure for the most complex GFW dataset (0.751) is achieved by the proposed model. This value is slightly higher than the best BCubed *F*-measure (0.745) reported in the original paper (*He et al., 2017*). However, the most important advantages of the proposed model from the practical point of view are excellent run-time/space complexity. For example, the inference in the proposed model is 5–10-times faster when compared to the VGGFace and VGGFace2. Moreover, the dimensionality of the feature vector is two to four times lower leading to the faster computation of a distance matrix in a clustering method. In addition, the proposed model makes it possible to simultaneously predict age and gender of observed facial image.

**Table 2 Clustering results, Gallagher dataset (C = 32 subjects).**

| | | *K/C* | ARI | AMI | Homogeneity | Completeness | *F*-measure |
|---|---|---|---|---|---|---|---|
| Single | VGGFace | 9.13 | 0.601 | 0.435 | 0.966 | 0.555 | 0.662 |
| | VGGFace2 | 2.75 | 0.270 | 0.488 | 0.554 | 0.778 | 0.637 |
| | Proposed model | 12.84 | 0.398 | 0.298 | **1.000** | 0.463 | 0.482 |
| Average | VGGFace | 1.84 | 0.858 | 0.792 | 0.916 | 0.817 | 0.874 |
| | VGGFace2 | 2.94 | 0.845 | 0.742 | 0.969 | 0.778 | 0.869 |
| | Proposed model | 2.03 | **0.890** | **0.809** | 0.962 | 0.832 | **0.897** |
| Complete | VGGFace | 1.31 | 0.571 | 0.624 | 0.886 | 0.663 | 0.706 |
| | VGGFace2 | 0.94 | 0.816 | 0.855 | 0.890 | **0.869** | 0.868 |
| | Proposed model | 1.47 | 0.644 | 0.649 | 0.921 | 0.687 | 0.719 |
| Weighted | VGGFace | **0.97** | 0.782 | 0.775 | 0.795 | 0.839 | 0.838 |
| | VGGFace2 | 1.63 | 0.607 | 0.730 | 0.876 | 0.760 | 0.763 |
| | Proposed model | 1.88 | 0.676 | 0.701 | 0.952 | 0.735 | 0.774 |
| Median | VGGFace | 9.16 | 0.613 | 0.433 | 0.942 | 0.555 | 0.663 |
| | VGGFace2 | 4.41 | 0.844 | 0.715 | 0.948 | 0.761 | 0.860 |
| | Proposed model | 12.38 | 0.439 | 0.324 | 0.960 | 0.482 | 0.531 |
| Rank-order | VGGFace | 1.59 | 0.616 | 0.488 | 0.902 | 0.582 | 0.702 |
| | VGGFace2 | 1.94 | 0.605 | 0.463 | 0.961 | 0.566 | 0.682 |
| | Proposed model | 3.06 | 0.249 | 0.251 | 0.986 | 0.424 | 0.398 |

**Note:**
The best results in each column are marked in bold.

**Table 3 Clustering results, GFW dataset (in average, C = 46 subjects).**

| | | *K/C* | ARI | AMI | Homogeneity | Completeness | *F*-measure |
|---|---|---|---|---|---|---|---|
| Single | VGGFace | 4.10 | 0.440 | 0.419 | 0.912 | 0.647 | 0.616 |
| | VGGFace2 | 3.21 | 0.580 | 0.544 | 0.942 | 0.709 | 0.707 |
| | Proposed model | 4.19 | 0.492 | 0.441 | 0.961 | 0.655 | 0.636 |
| Average | VGGFace | 1.42 | 0.565 | 0.632 | 0.860 | 0.751 | 0.713 |
| | VGGFace2 | 1.59 | 0.603 | **0.663** | 0.934 | 0.761 | 0.746 |
| | Proposed model | 1.59 | **0.609** | 0.658 | 0.917 | **0.762** | **0.751** |
| Complete | VGGFace | **0.95** | 0.376 | 0.553 | 0.811 | 0.690 | 0.595 |
| | VGGFace2 | 1.44 | 0.392 | 0.570 | 0.916 | 0.696 | 0.641 |
| | Proposed model | 1.28 | 0.381 | 0.564 | 0.886 | 0.693 | 0.626 |
| Weighted | VGGFace | 1.20 | 0.464 | 0.597 | 0.839 | 0.726 | 0.662 |
| | VGGFace2 | 1.05 | 0.536 | 0.656 | 0.867 | **0.762** | 0.710 |
| | Proposed model | 1.57 | 0.487 | 0.612 | 0.915 | 0.727 | 0.697 |
| Median | VGGFace | 5.30 | 0.309 | 0.307 | 0.929 | 0.587 | 0.516 |
| | VGGFace2 | 4.20 | 0.412 | 0.422 | 0.929 | 0.639 | 0.742 |
| | Proposed model | 6.86 | 0.220 | 0.222 | **0.994** | 0.552 | 0.411 |
| Rank-order | VGGFace | 0.82 | 0.319 | 0.430 | 0.650 | 0.694 | 0.630 |
| | VGGFace2 | 1.53 | 0.367 | 0.471 | 0.937 | 0.649 | 0.641 |
| | Proposed model | 1.26 | 0.379 | 0.483 | 0.914 | 0.658 | 0.652 |

**Note:**
The best results in each column are marked in bold.

**Table 4 Rank-1 accuracy (%) in face identification for LFW dataset.**

| ConvNet | Rank-1 accuracy |
|---|---|
| COTS-s1+s4 (*Best-Rowden et al., 2014*) | 66.5 |
| DeepFace (*Taigman et al., 2014*) | 64.9 |
| VGGFace (VGGNet-16) (*Parkhi, Vedaldi & Zisserman, 2015*) | 87.35 |
| ArcFace (*Deng et al., 2019*) | 92.56 |
| VIPLFaceNetFull (*Liu et al., 2017a*) | 92.79 |
| Light CNN (C) (*Wu et al., 2018*) | 93.8 |
| DeepID2+ (*Sun et al., 2014*) | 95.0 |
| DeepID3 (*Sun et al., 2015*) | 96.0 |
| IDL ensemble (*Liu et al., 2015*) | 98.03 |
| FaceNet (InceptionResNet+softmax loss) (*Schroff, Kalenichenko & Philbin, 2015*) | 97.72 |
| VGGFace2 (ResNet-50) (*Cao et al., 2018*) | 98.78 |
| Proposed model | 94.81 |

**Table 5 Results of face identification for PubFig83 dataset.**

| ConvNet | Rank-1 accuracy (%) | | | Model size (MB) | Inference time (ms) |
|---|---|---|---|---|---|
| | Number of photos per subject | | | | |
| | 5 | 15 | 30 | | |
| Light CNN (C) | 88.5 ± 0.9 | 92.9 ± 0.9 | 94.5 ± 0.2 | 21.9 | 39.6 |
| VGGFace (VGGNet-16) | 87.2 ± 1.0 | 94.5 ± 0.1 | 95.6 ± 0.0 | 552.6 | 186.8 |
| VGGFace2 (ResNet-50) | 97.0 ± 0.2 | 98.8 ± 0.0 | 99.0 ± 0.1 | 89.9 | 112.1 |
| Proposed model | 89.3 ± 0.8 | 96.2 ± 0.1 | 97.5 ± 0.1 | 13.5 | 29.1 |

Though the main purpose of this paper is face grouping in unsupervised environment, it is important to analyze the quality of face identification of the proposed model. Thus, in the last experiment of this subsection I deal with the LFW dataset (*Learned-Miller et al., 2016*) and PubFig83 dataset (*Pinto et al., 2011*). I took 9,164 photos of 1,680 persons from LFW with more than one photo. All 13,811 photos of $C = 83$ identities from PubFig83 were considered. The datasets are divided randomly 10 times into the training and testing sets, so that the ratio of the size $R$ of the training set to the size of the whole dataset is equal to a fixed number. The rank-1 accuracy of the k-NN classifier for the proposed model in comparison with the state-of-the-art results for 50–50 train/test split of the LFW is shown in Table 4. The results of the 10-times repeated random sub-sampling cross validation of the k-NN classifier (*Savchenko, 2018*) for several publicly available ConvNets in dependence of the number of photos of each subject are presented in Table 5. Here the model size of each ConvNet and average inference time on CPU of MacBook Pro 2015 (16 GB RAM, Intel Core i7 2.2 GHz) are additionally presented.

Here one could notice that the proposed simple network provides competitive results with the state-of-the-art solutions though its size is approximately 6.5-times lower than the

size of the deep ResNet-50 trained on VGGFace2 dataset (*Cao et al., 2018*). The inference time in the MobileNet is also known to be much lower.

## Age and gender recognition

In this subsection, the proposed model is compared with publicly available ConvNets for age/gender prediction:

- Deep expectation (DEX) VGG16 network trained on the IMDB-Wiki dataset (*Rothe, Timofte & Van Gool, 2015*)
- Wide ResNet (*Zagoruyko & Komodakis, 2016*) (weights.28-3.73) (https://github.com/yu4u/age-gender-estimation)
- Wide ResNet (weights.18-4.06) (https://github.com/Tony607/Keras_age_gender)
- FaceNet (*Schroff, Kalenichenko & Philbin, 2015*) (https://github.com/BoyuanJiang/Age-Gender-Estimate-TF)
- BKNetStyle2 (https://github.com/truongnmt/multi-task-learning)
- SSRNet (*Yang et al., 2018*) (https://github.com/shamangary/SSR-Net)
- MobileNet v2 (Agegendernet) (https://github.com/dandynaufaldi/Agendernet)
- Two models from InsightFace (*Deng et al., 2019*): original ResNet-50 and "new" fast ConvNet (https://github.com/deepinsight/InsightFace/)
- Inception v3 (https://github.com/dpressel/rude-carnie) fine-tuned on the Adience dataset (*Eidinger, Enbar & Hassner, 2014*)
- Age_net/gender_net (*Levi & Hassner, 2015*) trained on the Adience dataset (*Eidinger, Enbar & Hassner, 2014*).

In contrast to the proposed approach, all these models have been trained only on specific datasets with age and gender labels, that is, they are fine-tuned from traditional ConvNets pre-trained on ImageNet-1000 and do not use large-scale face recognition datasets.

In addition, several special cases of the MobileNet-based model (Fig. 1) were examined. Firstly, I compressed this model using standard Tensorflow quantization graph transforms. Secondly, *all* layers of the proposed model were fine-tuned for age and gender predictions (hereinafter "Proposed MobileNet, fine-tuned"). Though such tuning obviously reduce the accuracy of face identification with identity features at the output of the base MobileNet, it caused an above-mentioned increase of validation accuracies for gender and age classification Thirdly, in order to compare the proposed multi-output neural network (Fig. 1) with conventional approach, I additionally used the same MobileNet-based network but with a single head, which was pre-trained on the same VGGFace2 face recognition problem and then fine-tuned for one task (age or gender recognition), that is, there are two different models (hereinafter "MobileNet with single head") for all these tasks. Finally, it was decided to measure the influence of pre-training on face recognition task. Hence, the model (Fig. 1) was fine-tuned using the above-mentioned procedure and the same training set with labeled age and gender, but the base

**Table 6 Performance analysis of ConvNets.**

| ConvNet | Model size (MB) | Average CPU inference time (ms) | | |
| --- | --- | --- | --- | --- |
| | | Laptop | Mobile phone 1 | Mobile phone 2 |
| age_net/gender_net | 43.75 | 91 | 1,082 | 224 |
| DEX | 513.82 | 210 | 2,730 | 745 |
| Proposed MobileNet | 13.78 | 21 | 354 | 69 |
| Proposed MobileNet, quantized | 3.41 | 19 | 388 | 61 |

MobileNet was pre-trained on conventional ImageNet-1000 dataset rather than on VGGFace2 (*Cao et al., 2018*). Though such network (hereinafter "Proposed MobileNet, fine-tuned from ImageNet") cannot be used in facial clustering, it can be applied in gender and age prediction tasks.

I run the experiments on the MSI GP63 8RE laptop (CPU: 4xIntel Core i7 2.2 GHz, RAM: 16 GB, GPU: nVidia GeForce GTX 1060) and two mobile phones, namely: (1) Honor 6C Pro (CPU: MT6750 4×1 GHz and 4×2.5 GHz, RAM: 3 GB); and (2) Samsung S9+ (CPU: 4×2.7 GHz Mongoose M3 and 4×1.8 GHz Cortex-A55, RAM: 6 GB). The size of the model file and average inference time for one facial image are presented in Table 6.

As expected, the MobileNets are several times faster than the deeper convolutional networks and require less memory to store their weights. Though the quantization reduces the model size in four times, it does not decrease the inference time. Finally, though the computing time for the laptop is significantly lower when compared to the inference on mobile phones, their modern models ("Mobile phone 2") became all the more suitable for offline image recognition. In fact, the proposed model requires only 60–70 ms to extract facial identity features and predict both age and gender, which makes it possible to run complex analytics of facial albums on device.

In the next experiments the accuracy of the proposed models were estimated in gender recognition and age prediction tasks. At first, I deal with University of Tennessee, Knoxville Face Dataset (UTKFace) (*Zhang, Song & Qi, 2017*). The images from complete ("In the Wild") set were pre-processed using the following procedure from the above-mentioned Agegendernet resource (https://github.com/dandynaufaldi/Agendernet): faces are detected and aligned with margin 0.4 using get_face_chip function from DLib. Next, all images which has no single face detected, are removed. The rest 23,060 images are scaled to the same size 224×224. In order to estimate the accuracy of age prediction, eight age ranges from the Adience dataset (*Eidinger, Enbar & Hassner, 2014*) were used. If the predicted and real age are included into the same range, then the recognition is assumed to be correct. The results are shown in Table 7. In contrast to the previous experiment (Table 6), here the inference time is measured on the laptop's GPU.

In this experiment the proposed ConvNets (three last lines in Table 7) have higher accuracy of age/gender recognition when compared to the available models trained only on specific datasets, for example, IMDB-Wiki or Adience. For example, the best fine-tuned MobileNet provided 2.5–40% higher accuracy of gender classification. The gain in age prediction performance is even more noticeable: I obtain 1.5–10 years less MAE (mean

**Table 7 Age and gender recognition results for preprocessed UTKFace (in the wild) dataset.**

| Models | Gender accuracy (%) | Age MAE | Age accuracy (%) | Model size (Mb) | Inference time (ms) |
|---|---|---|---|---|---|
| DEX | 91.05 | 6.48 | 51.77 | 1,050.5 | 47.1 |
| Wide ResNet (weights.28-3.73) | 88.12 | 9.07 | 46.27 | 191.2 | 10.5 |
| Wide ResNet (weights.18-4.06) | 85.35 | 10.05 | 43.23 | 191.2 | 10.5 |
| FaceNet | 89.54 | 8.58 | 49.02 | 89.1 | 20.3 |
| BKNetStyle2 | 57.76 | 15.94 | 23.49 | 29.1 | 12.5 |
| SSRNet | 85.69 | 11.90 | 34.86 | 0.6 | 6.6 |
| MobileNet v2 (Agegendernet) | 91.47 | 7.29 | 53.30 | 28.4 | 11.4 |
| ResNet-50 from InsightFace | 87.52 | 8.57 | 48.92 | 240.7 | 25.3 |
| "New" model from InsightFace | 84.69 | 8.44 | 48.41 | 1.1 | 5.1 |
| Inception trained on Adience | 71.77 | – | 32.09 | 85.4 | 37.7 |
| age_net/gender_net | 87.32 | – | 45.07 | 87.5 | 8.6 |
| MobileNets with single head | 93.59 | 5.94 | 60.29 | 25.7 | 7.2 |
| Proposed MobileNet, fine-tuned from ImageNet | 91.81 | 5.88 | 58.47 | 13.8 | 4.7 |
| Proposed MobileNet, pre-trained on VGGFace2 | 93.79 | 5.74 | 62.67 | 13.8 | 4.7 |
| Proposed MobileNet, fine-tuned | 94.10 | 5.44 | 63.97 | 13.8 | 4.7 |

absolute error) and 10–40% higher accuracy. Though the gender recognition accuracy of a ConvNet with single head is rather high, multi-task learning causes a noticeable decrease in age prediction quality (up to 0.5% and 4.5% differences in MAE accuracy). Hence, the proposed approach is definitely more preferable if both age and gender recognition tasks are solved due to the twice-lower running time when compared to the usage of separate age and gender networks. It is interesting that though there exist models with lower size, for example, SSRNet (*Yang et al., 2018*) or new InsightFace-based model (*Deng et al., 2019*), the proposed ConvNet provides the fastest inference, which can be explained by special optimization of hardware for such widely used architectures as MobileNet.

It is known that various image preprocessing algorithms could drastically influence the recognition performance. Hence, in the next experiment conventional set of all 23,708 images from "Aligned & cropped faces" provided by the authors of the UTKFace was used. The results of the same models are presented in Table 8.

The difference in image pre-processing causes significant *decrease* of the accuracies of most models. For example, the best proposed model in this experiment has 14–40% and 5–40% higher age and gender recognition accuracy, respectively. Its age prediction MAE is at least 2.5 years lower when compared to the best publicly available model from Insight face. The usage of the same DLib library to detect and align faces in a way, which is only slightly different from the above-mentioned preprocessing pipeline, drastically decreases the accuracy of the best existing model from previous experiment (MobileNet v2) up to 5.5% gender accuracy and 3 years in age prediction MAE. Obviously, such dependence of performance on the image pre-processing algorithm makes the model inappropriate for practical applications. Hence, this experiment clearly demonstrates how

**Table 8 Age and gender recognition results for UTKFace (aligned and cropped faces) dataset.**

| Models | Gender accuracy (%) | Age MAE | Age accuracy (%) |
|---|---|---|---|
| DEX | 83.16 | 9.84 | 41.22 |
| Wide ResNet (weights.28-3.73) | 73.01 | 14.07 | 29.32 |
| Wide ResNet (weights.18-4.06) | 69.34 | 13.57 | 37.23 |
| FaceNet | 86.14 | 9.60 | 44.70 |
| BKNetStyle2 | 60.93 | 15.36 | 21.63 |
| SSRNet | 72.29 | 14.18 | 30.56 |
| MobileNet v2 (Agegendernet) | 86.12 | 11.21 | 42.02 |
| ResNet-50 from InsightFace | 81.15 | 9.53 | 45.30 |
| "New" model from InsightFace | 80.55 | 8.51 | 48.53 |
| Inception trained on Adience | 65.89 | – | 27.01 |
| age_net/gender_net | 82.36 | – | 34.18 |
| MobileNets with single head | 91.89 | 6.73 | 57.21 |
| Proposed MobileNet, fine-tuned from ImageNet | 84.30 | 7.24 | 58.05 |
| Proposed MobileNet, pre-trained on VGGFace2 | 91.95 | 6.00 | 61.70 |
| Proposed MobileNet, fine-tuned | 91.95 | 5.96 | 62.74 |

the proposed model exploits the potential of very large face recognition dataset to learn face representation in contrast to the training only on rather small and dirty datasets with age and gender labels. It is important to emphasize that the same statement is valid even for the proposed model (Fig. 1). In particular, the usage of face identification features pre-trained on VGGFace2 leads to 3.5% and 6.5% lower error rate of age and gender classification, respectively, when compared to conventional fine-tuning of MobileNet, which has been preliminarily trained on ImageNet-1000 only (third last line in Table 8). This difference in error rates is much more noticeable when compared to the previous experiment (Table 7), especially for age prediction MAE.

Many recent papers devoted to UTKFace dataset split it into the training and testing sets and fine-tune the models on such training set (*Das, Dantcheva & Bremond, 2018*). Though the proposed ConvNet does not require such fine-tuning, its results are still very competitive. For example, I used the testing set described in the paper (*Cao, Mirjalili & Raschka, 2019*), which achieves the-state-of-the-art results on a subset of UTKFace if only 3,287 photos of persons from the age ranges are taken into the testing set. The proposed model achieves 97.5% gender recognition accuracy and age prediction MAE 5.39. It is lower than 5.47 MAE of the best CORAL-CNN model from this paper, which was additionally trained on other subset of UTKFace.

As the age and gender recognition is performed in the proposed pipeline (Fig. 2) for a *set* of facial images in a cluster, it was decided to use the known video datasets with age/gender labels in the next experiments in order to test performance of classification of a set of video frames (*Kharchevnikova & Savchenko, 2018*):

- Eurecom Kinect (*Min, Kose & Dugelay, 2014*), which contains nine photos for each of 52 subjects (14 women and 38 men).

- Indian movie face database (IMFDB) (*Setty et al., 2013*) with 332 video clips of 63 males and 33 females. Only four age categories are available: "Child" (0–15 years old), "Young" (16–35), "Middle": (36–60), and "Old" (60+).
- Acted facial expressions in the wild (AFEW) from the EmotiW 2018 (Emotions recognition in the wild) audio–video emotional sub-challenge (*Klare et al., 2015*). It contains 1,165 video files. The facial regions were detected using the MTCNN (*Zhang et al., 2016a*).
- IARPA Janus Benchmark A (*Dhall et al., 2012*) with more than 13,000 total frames of 1,165 video tracks. Only gender information is available in this dataset.

In video-based gender recognition a gender of a person on each video frame is firstly classified. After that two simple fusion strategies are utilized, namely, simple voting, and the product rule (2). The obtained accuracies of the proposed models compared to most widely used DEX (*Rothe, Timofte & Van Gool, 2015*) and gender_net/age_net (*Levi & Hassner, 2015*) are shown in Table 9.

Here again the proposed models are much more accurate than the publicly available ConvNets trained only on rather small and dirty datasets with age/gender labels. It is important to emphasize that the gender output of the proposed network was trained on the same IMDB-Wiki dataset as the DEX network (*Rothe, Timofte & Van Gool, 2015*). However, the error rate in the proposed approach is much lower when compared to the DEX. It can be explained by the pre-training of the base MobileNet on face identification task with very large dataset, which helps to learn rather good facial representations. Such pre-training differs from traditional usage of ImageNet weights and only fine-tune the CNN on a specific dataset with known age and gender labels. Secondly, the usage of product rule generally leads to 1–2% decrease of the error rate when compared to the simple voting. Thirdly, the fine-tuned version of the proposed model achieves the lowest error rate only for the Kinect dataset and is 1–3% *less* accurate in other cases. It is worth noting that the best accuracy for Eurecom Kinect dataset is 7% higher than the best-known accuracy (87.82%) achieved by *Huynh, Min & Dugelay (2012)* in similar settings without analysis of depth images. Fourthly, though the compression of the ConvNet makes it possible to drastically reduce the model size (Table 6), it is characterized by up to 7% decrease of the recognition rate. Finally, conventional single-output model is slightly less accurate than the proposed network, though the difference in the error rate is not statistically significant.

In the last experiment the results for age predictions are presented (Table 10). As the training set for the proposed network differs with conventional DEX model due to addition of the Adience data to the IMDB-Wiki dataset, it was decided to repeat training of the proposed network (Fig. 1) with the IMDB-Wiki data only. Hence, the resulted "Proposed MobileNet, IMDB-Wiki only" ConvNet is more fairly compared with the DEX network.

Here it was assumed that age is recognized correctly for the Kinect and AFEW datasets (with known age) if the difference between real and predicted age is not greater than 5 years. The fusion of age predictions of individual video frames is implemented

**Table 9 Video-based gender recognition accuracy, %.**

| ConvNet | Aggregation | Eurecom Kinect | IMFDB | AFEW | IJB-A |
|---|---|---|---|---|---|
| gender_net | Simple voting | 73 | 71 | 75 | 60 |
| | Product rule | 77 | 75 | 75 | 59 |
| DEX | Simple Voting | 84 | 81 | 80 | 81 |
| | Product rule | 84 | 88 | 81 | 82 |
| MobileNet with single head | Simple voting | 93 | 97 | 92 | 95 |
| | Product rule | 93 | 98 | **93** | 95 |
| Proposed MobileNet | Simple voting | 94 | 98 | **93** | 95 |
| | Product rule | 93 | **99** | **93** | **96** |
| Proposed MobileNet, quantized | Simple voting | 88 | 96 | 92 | 93 |
| | Product rule | 86 | 96 | 93 | 94 |
| Proposed MobileNet, fine-tuned | Simple voting | 93 | 95 | 91 | 94 |
| | Product rule | **95** | 97 | 92 | 95 |

Note:
The highest accuracies for each dataset are marked by bold.

**Table 10 Video-based age prediction accuracy, %.**

| ConvNet | Aggregation | Eurecom Kinect | IMFDB | AFEW |
|---|---|---|---|---|
| age_net | Simple voting | 41 | 68 | 27 |
| | Product rule | 45 | 48 | 27 |
| | Expected value | 69 | 32 | 30 |
| DEX | Simple voting | 60 | 29 | 47 |
| | Product rule | 71 | 29 | 48 |
| | Expected value | 71 | 54 | 52 |
| MobileNet with single head | Simple voting | 91 | 34 | 46 |
| | Product rule | 93 | 38 | 47 |
| | Expected value | **94** | 75 | **54** |
| Proposed MobileNet, IMDB-Wiki only | Simple voting | 73 | 30 | 47 |
| | Product rule | 83 | 31 | 47 |
| | Expected value | 85 | 58 | 52 |
| Proposed MobileNet, IMDB-Wiki+Adience | Simple voting | 92 | 32 | 46 |
| | Product rule | **94** | 36 | 46 |
| | Expected value | **94** | **77** | **54** |
| Proposed MobileNet, quantized | Simple voting | 86 | 34 | 44 |
| | Product rule | 88 | 36 | 46 |
| | Expected value | 85 | 58 | 50 |
| Proposed MobileNet, fine-tuned | Simple voting | 74 | 33 | 45 |
| | Product rule | 77 | 35 | 45 |
| | Expected value | 92 | 72 | 51 |

Note:
The highest accuracies for each dataset are marked by bold.

by: (1) simple voting, (2) maximizing the product of age posterior probabilities (2), and (3) averaging the expected value (3) with choice of $L = 3$ top predictions in each frame.

One can notice that the proposed models are again the most accurate in practically all cases. The accuracy of the DEX models are comparable with the proposed ConvNets only for the AFEW dataset. This gain in the error rate cannot be explained by using additional Adience data, as it is noticed even for the "Proposed MobileNet, IMDB-Wiki only" model. Secondly, the lowest error rates are obtained for the computation of the expected value of age predictions. For example, it is 2% and 8% more accurate than the simple voting for the Kinect and AFEW data. The effect is especially clear for the IMFDB images, in which the expected value leads to up to 45% higher recognition rate.

## CONCLUSIONS

In this paper I proposed an approach to organizing photo and video albums (Fig. 2) based on a simple extension of MobileNet (Fig. 1), in which the facial representations suitable for face identification, age, and gender recognition problems are extracted. The main advantage of the proposed model is the possibility to solve all three tasks simultaneously without need for additional ConvNets. As a result, a very fast facial analytic system was implemented (Table 6), which can be installed even on mobile devices (Fig. 3). It was shown that the proposed approach extracts facial clusters rather accurately when compared to the known models (Tables 1 and 2). Moreover, the known state-of-the-art BCubed $F$-measure for very complex GFW data was slightly improved (Table 3). More importantly, the results for age and gender predictions using extracted facial representations significantly outperform the results of the publicly available neural networks (Tables 9 and 10). The state-of-the-art results on the whole UTKFace dataset (*Zhang, Song & Qi, 2017*) was achieved (94.1% gender recognition accuracy, 5.44 age prediction MAE) for the ConvNets which are not fine-tuned on a part of this dataset.

The proposed approach does not have the limitations of existing methods of simultaneous face analysis (*Ranjan, Patel & Chellappa, 2017*; *Yoo et al., 2018*) for usage in face identification and clustering tasks because it firstly learns the facial representations using external very large face recognition dataset. The proposed approach is usable even for face identification with small training samples, including the most complex case, namely, a single photo per individual. Furthermore, the method enables to apply the ConvNet in completely unsupervised environments for face clustering, given only a set of photos from a mobile device. Finally, the training procedure to learn parameters of the method alternately trains all the facial attribute recognition tasks using batches of different training images. Hence, the training images are not required to have all attributes available. As a result, much more complex (but still fast) network architectures can be trained when compared to the ConvNet of (*Yoo et al., 2018*) and, hence, achieve very high age/gender recognition accuracy and the face clustering quality comparable to very deep state-of-the-art ConvNets.

In future works it is necessary to deal with the aging problem. In fact, the average linkage clustering usually produces several groups for the same person (especially, a child).

The single linkage clustering can resolve this issue if there are photos of the same subject made over the years. Unfortunately, the performance of the single linkage is rather poor when compared to another agglomerative clustering methods (Tables 1–3). An additional research direction is a thorough analysis of distance measures in the facial clustering (Zhu, Wen & Sun, 2011); that is, the usage of distance learning (Zhu et al., 2013) or special regularizers (Savchenko & Belova, 2018). It is also important to extend the proposed approach to classify other facial attributes, for example, race/ethnicity (Zhang, Song & Qi, 2017) or emotions (Rassadin, Gruzdev & Savchenko, 2017). Finally, though the main purpose of this paper was to provide an efficient ConvNet suitable for multiple tasks including extraction of good face representations, the quality of grouping faces could be obviously improved by replacement of agglomerative clustering to contemporary clustering techniques with unknown number of clusters (He et al., 2017; Zhang et al., 2016b; Shi, Otto & Jain, 2018; Vascon et al., 2013).

### Funding

This research was supported by Samsung Research and Samsung Electronics. Additionally, this research was prepared within the framework of the Basic Research Program at the National Research University Higher School of Economics (HSE) in return for lab time. There was no additional external funding received for this study. The funders had no role in study design, data collection and analysis, decision to publish, or preparation of the manuscript.

### Grant Disclosures

The following grant information was disclosed by the authors:
Samsung Research and Samsung Electronics.
Basic Research Program at the National Research University Higher School of Economics.

### Competing Interests

Andrey V. Savchenko is an employee of the Samsung-PDMI Joint AI Center, St. Petersburg Department of Steklov Institute of Mathematics Russian Academy of Sciences and of the National Research University Higher School of Economics.

### Author Contributions

- Andrey V. Savchenko conceived and designed the experiments, performed the experiments, analyzed the data, contributed reagents/materials/analysis tools, prepared figures and/or tables, performed the computation work, authored or reviewed drafts of the paper, approved the final draft.

### Data Availability

Source code and neural network models are publicly available at GitHub:
https://github.com/HSE-asavchenko/HSE_FaceRec_tf/tree/master/age_gender_identity.

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
