# Peer review of "Efficient facial representations for age, gender and identity recognition in organizing photo albums using multi-output ConvNet"

_PeerJ Computer Science, doi:10.7717/peerj-cs.197_

## Round 0.1 · original submission · Major Revisions

Please consider the comments of the reviewers and prepare a revision of the paper.

Reviewer 1 ·

Basic reporting

The quality of English language is high. The text is written correctly with minor mistakes:
1) "that not all images ... contains information" (188 line)
2) "can be applied is we naively assume" (216 line)

Experimental design

The proposed CNN and clustering pipeline show the novelty of the approach. Everything is described clear and with maximum details. The description provided make the paper easy to replicate to test the approach. All the methods sections and subsections leave good impression about the research overall.

Validity of the findings

The experimental part is written very detailed and the authors obtained high precision/recall results in comparison to existing approaches. Moreover, the proposed MobileNet-based CNN lead to low computational complexity and high inference speed on mobile devices. What I would like to recommend is to do research on a new dataset UTKFaces, which contains information about age/gender/ethnicity. Seems the proposed Mobilenet structure can be extended to the 3rd output for ethnicity detection and this will be a good improvement in the research field.

Reviewer 2 ·

Basic reporting

The paper describes a single lightweight CNN model to handle face identification, age and gender recognition problems. The proposed method trains only data relevant part of the network by alternating the mini-batches with age and gender info. The experimental results show that the proposed model has impressive run-time/space complexity value. The related work section also appears to be strong.

On the other hand, the paper is a bit difficult to follow, the language and paper organization can ideally be improved. To give an example, some information about experiments are shared in the "Multi-Output ConvNet for Simultaneous Age, Gender and Identity Recognition" section. Moreover, there are some claims (e.g. line 199-201) without an experiment or external scientific reference. Also, I do not understand the sentence in lines 225. Finally, there are typos such as simulatenous->simultaneous in line 140.

Experimental design

The experiments give some insight about the strengths and weaknesses of the proposed method. The experimental results show that the proposed model obtained promising results in terms of model size and accuracy.

Some details about the model are given in the shared code link. To give an example, information about loss function and learning rate are not shared in manuscript or missed it. It would be nice to see these information in the manuscript to reproduce same experiments.

Validity of the findings

The paper looks technically sound and the problem is well motivated. Moreover, the proposed approach is simple and reasonable. However, there are some points that need explanation, respectively:

- According to the Figure 2, identity features and born year predictions are given as input to the clustering operation. However, different situation is mentioned in the line 209-210.

- Where is the proof of the claim in lines 202, 203 and 204?

- How do you encode age values? What is the loss function for it?

- Do you have joint loss function for learning gender and age probabilities?

Additional comments

My general observation about the manuscript is that idea is nice, but text organization is very poor.

Reviewer 3 ·

Basic reporting

I think this is an interesting paper on a relevant topic.
The language is clear and informative.
The author provides all the necessary references. Many of them are dated to the last few years that confirms the awareness of the state-of the art in the field.
The paper is well-structured. All the figures and tables are relevant and discussed in the paper in details.
However, the paper contains some typos and minor issues, which are listed in “General comments for the author” section.

Experimental design

The paper relates to the most popular computer science fields, namely, machine learning and mobile development.
More precisely, the article deals with the actual problem of recognizing gender, age and person identification in digital images.
One of the most valuable pros. of the paper is the number of experiments. The experiments are dedicated to choosing the best algorithms and comparison with the available models.
The proposed technique is well described. The implementation is available at github for the community.

Validity of the findings

I think that the work is significant and will be useful to specialists in the field.

Additional comments

- I think it would be better to state in abstract that the proposed technique is ready to use with mobile devices.
- Please, clarify why dataset «contains incorrect ground truth values of age» (line 45)
- MTCNN is used without reference (line 208)
- Please, clarify what “most important” means in «Moreover, we assume that the most important clusters should not contain photos/videos made in one day» (line 244-245)
- Please, check if the complexity should be higher: “computational complexity of such clustering is 3-4-times lower” (line 302)
- Please, check if the “inference” should be “faster” or the time should be “lower”: “The inference time in the MobileNet is also known to be much slower.” (line 328-329)
- Please, check line thickness in fig. 2as the bottom dash line disappears at some (ordinary)) scale in my Acrobat Reader.
Typos:
- «have been study» 81
- «resulted model are made»267
Missed or wrong prepositions:
- «using embeddings at one the last layers» (line 77)
- «using embeddings at one the last layers» (line 78)
- «information interesting a target group» (line 100)
- “to predict in an observed person is 1,2,... or 100 years old”
Missed verb: «At first, the base MobileNet for face identification using very large facial dataset, e.g., VGGFace2” line 259-260)

---

## Round 0.2 · accepted · Accept

The reviewers agree that the revision addresses their comments. The revised manuscript can be accepted for publication

Congratulations

Reviewer 1 ·

Basic reporting

The new version seems much better and the updated experiments provide better conclusion of the research.

Experimental design

Experiments added on other datasets like UTKFaces gave a better understanding of the value of conducted research.

Validity of the findings

No comment

Additional comments

The paper presents a concluded research now and the text is filled with different experiment scenarios and results. I have no more comments on the paper

Reviewer 2 ·

Basic reporting

The sections in the manuscript seem to be correctly separated, after revisions.

Experimental design

The additional experiments respond to my previous questions. Moreover, rebuttal text advocate the arguments put forward in the manuscript more strongly by using these experiments.

Validity of the findings

The method used looks technically sound.

Additional comments

Thank you very much for your corrections and responses. The proposed rebuttal and experiments answer my questions. My views are to accept the manuscript.

Reviewer 3 ·

Basic reporting

I think this is an interesting paper on a relevant topic.
The language is clear and informative.
The author provides all the necessary references. Many of them are dated to the last few years that confirms the awareness of the state-of the art in the field.
The paper is well-structured. All the figures and tables are relevant and discussed in the paper in details.
The revised version of the paper contains all the necessary corrections and additions.

Experimental design

The paper relates to the most popular computer science fields, namely, machine learning and mobile development.
More precisely, the article deals with the actual problem of recognizing gender, age and person identification in digital images.
One of the most valuable pros. of the paper is the number of experiments. The experiments are dedicated to choosing the best algorithms and comparison with the available models.
The proposed technique is well described. The implementation is available at github for the community.

Validity of the findings

I think that the work is significant and will be useful to specialists in the field.

Additional comments

As the revised version of the paper contains all the necessary corrections and additions, I vote for accepting this paper.